# metaSVG: A Portable Exchange Format for Adaptable Laser Cutting Plans

Nur Yildirim*        Matthew Franklin        Daniel Zeng        John Zimmerman        James McCann†

Carnegie Mellon University

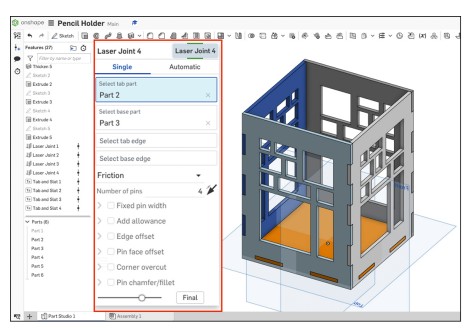
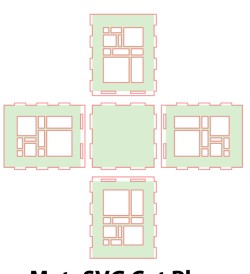

**MetaSVG Cut Plan**

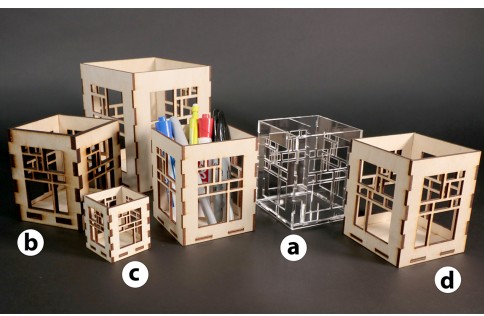

Figure 1: A pencil holder exported as metaSVG (middle) from CAD (left) can be automatically adapted for laser cutting out of (a) different materials, (b) material thicknesses, (c) scales, and (d) on different laser cutters.

## ABSTRACT

2D cut paths for laser cutters are specific to material parameters (type, thickness), design parameters (overall size, desired joint fit), and laser settings. Changing any of these parameters – because, e.g., one wants to cut a design from a new material, or on a different machine – requires by-hand adjustment of the cut path to compensate. This adjustment is complicated by the fact that 2D path formats lack information about how high-level parameters influenced the design, leaving makers to infer the original author's intent when adjusting paths. We present metaSVG, an exchange format that includes joint parameters as metadata when exporting cut plans from parametric 3D CAD. This allows our software tool, metaSVG Print, to automatically adapt metaSVG cut plans to new sizes, materials, material thickness, and laser cutters. We demonstrate the effectiveness of metaSVG by augmenting existing designs with metadata and fabricating them with different materials and laser cutters.

**Index Terms:** Applied computing—Computer-aided manufacturing——

## 1 INTRODUCTION

Digital fabrication promises shareable plans, plans that can be uploaded once by authors and downloaded and materialized effortlessly by makers all over the world. However, to meet this promise, fabrication plans must be stored in a portable format – a format that can be interpreted to produce the desired object by a wide range of printing systems. This is already true for 3D printers (STL files) and for 2D printers (PDF files), but it is not yet true for laser cutters. The problem is that SVG files (the common distribution format for laser cutting plans) do not contain enough information to adapt their contents to new laser cutters, materials, and scales.

The non-portability of laser cutting plans impacts both *authors* (people who create and share cut plans) and *makers* (people who download and laser cut the plan). For makers, the need to tweak every joint in a downloaded cut plan presents a major barrier [1, 13, 31]. Even for makers who possess the skills to manipulate cut

---

*yildirim@cmu.edu

†jmccann@cs.cmu.edu

plans, the design adaptation task is tedious, time consuming, and error prone [31]. For authors who create and share designs, the need to adapt every joint limits the impact of their design. This leads to less recognition, one of the main motivations authors have for freely sharing their plans [8, 13, 17]. As a workaround, authors sometimes provide detailed documentation and print specifications, which significantly increases the effort to create and share a plan [17, 41].

To address these challenges, we developed a workflow (Figure 1) around metaSVG: a file format that adds joint metadata into an SVG file in a backwards-compatible way. This metadata includes everything needed to recompute the cut plan for new materials, material thicknesses, scales, and laser cutters. MetaSVG stores part shapes, joint locations, joint types (i.e., box, tab-and-slot, slotted, t-slot joints), joint fits (i.e., clearance, location, interference), and joint angles. For authors, we created metaSVG Exporter, an extension for the OnShape 3D CAD system, that allows authors to encode this metadata when exporting SVG cut plans. For makers, we created metaSVG Print, which works like a print dialog box, allowing makers to specify the scale of their model along with material, thickness, and detailed calibration information for their specific laser cutter (i.e., kerf, joint fit allowances, and path format settings for output). These settings can be stored as presets (e.g., by a shop manager), making cut plan adaptation as simple as 2D document printing. Our workflow enables authors to share portable, reusable laser cut plans, and it allows makers to adapt cut plans to materials and laser cutters without having to make manual adjustments.

This paper documents our design and implementation of metaSVG, metaSVG Exporter, and metaSVG Print. It also provides an assessment of the current metaSVG workflow. We re-created and adapted (to different materials, thicknesses, scales, and laser cutters) a wide range of designs from Thingiverse. We also assessed the applicability of the metaSVG concept to laser-cuttable models on Thingiverse in general. Our research contributes:

- an accelerated workflow for authoring cut plans, adapting cut plans, and sharing laser cutters calibration information;

- metaSVG, a portable exchange format that augments the SVG format with material- and joint-specific metadata;

- metaSVG Exporter, an OnShape extension for authoring metaSVG cut plans when exporting from 3D CAD;

## Current Process

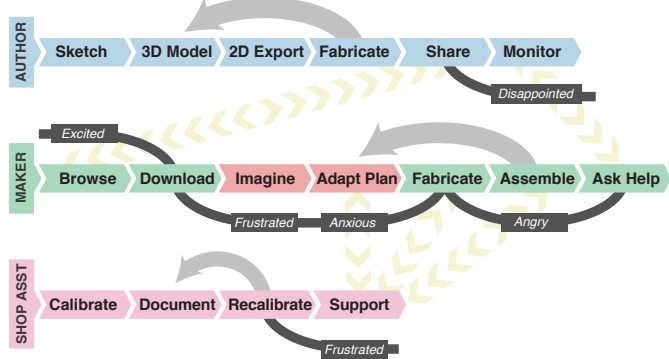

## metaSVG Process

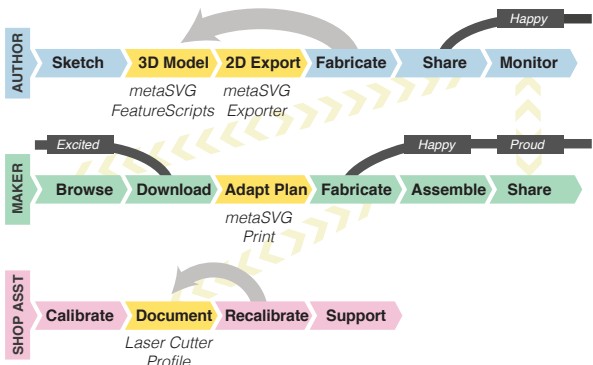

Figure 2: Journey maps of the current (left) and proposed (right) laser cutter workflows. Current workflows require manual material- and laser-cutter-dependent adjustment of designs, leading to tedious edit/test cycles, frustration, and disappointment. With metaSVG, plans can be adapted automatically by software, eliminating edit/test cycles and leading to – we hope – better outcomes.

- metaSVG Print, a browser-based application that can adapt metaSVG cut plans using design metadata and a laser cutter calibration profile; and

- a technical evaluation that demonstrates the effectiveness of our advance and reveals where it can be extended and improved.

## 2 MOTIVATION

The design, sharing, and fabrication of laser cutting plans typically involves three roles. Authors (often working in 3D CAD) make and share their designs as 2D cut plans. Makers download plans and recreate a design by cutting material on a laser cutter. Shop assistants care for laser cutters, and provide resources (e.g., cut settings for specific materials) and assistance to makers in shops, labs, and makerspaces [47]. A single person might enact all of these roles or they might only enact one of them.

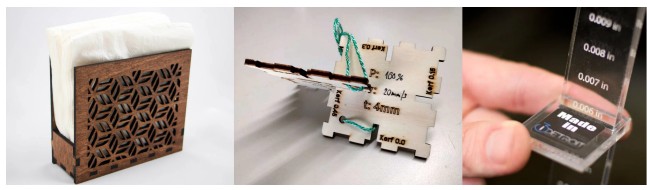

Figure 3: A laser cut napkin holder design (left), commonly used kerf gauges for calibrating laser cut joints (center, right).

We illustrate the challenges posed by using a non-portable cut plan format by considering the lifecycle of a napkin holder design (Figure 3, left). A prolific laser cutting author enjoys creating new designs to share via the internet. The author realizes that laser cutting would be well-suited to creating an elegant napkin holder, and that such a design might do well among makers because no other authors have created such a design. They, therefore, model the napkin holder in a 3D CAD tool, using existing parametric *features* (composite CAD tools that produce complex geometry from simple user inputs) to generate both box joints and tab-and-slot joints to connect the parts [39]. Once the holder looks good in CAD, they export an SVG, test on their laser cutter, and adjust the model until it works well on their laser cutter and target material (5mm plywood). After a few test-and-adjust cycles, the design fabricates cleanly on their laser cutter. The author takes a few photos of the finished build and uploads them along with the SVG cut plan to a model sharing web page.

A maker looking for a small home decor project discovers the napkin holder the next day when browsing for new models. She decides to make a version of the model using 3mm acrylic and the laser cutter at her local makerspace. The first thing the maker needs to do is adapt the SVG cutting plan to the new material thickness. She does this by opening the SVG file in Adobe Illustrator and moving each tab and slot part 1mm inward to account for the thinner acrylic as compared to the original plywood. This is especially difficult for joints that aren't perpendicular to the X or Y axis, but the maker is a skilled Illustrator user and is able to use line snapping and some additional paths to make the change.

Once the plan is adapted she walks to the makerspace to cut it, but finds that the resulting pieces have a very tight fit that cracks the material. Seeking the advice of the shop assistant, she learns that the laser cutter in the makerspace has a kerf (cut width) which is generally a bit smaller than most hobbyist cutters, so folks downloading plans from the internet often need to inset the paths slightly to compensate. The assistant is able to provide the maker with a recent kerf value measured on the shop's cutter (Figure 3, center, right), and the maker uses this value to inset the cut path before doing a second run of the plan. This time, the joints fit together but are a bit loose because plywood (the author's test material) and acrylic (the maker's chosen material) have different friction and compression behaviors. Not wanting to edit the plan further, the maker accepts these sloppy joints and glues the napkin holder together.

This workflow (Figure 2, left) serves none of the roles – the author has fewer successful builds of their plan, the maker uses extra time and material, and the shop assistant spends time debugging what should have been a straightforward build. With our new tools and improved workflow (Figure 2, right), the author is able to export a metaSVG file to share. This export works without any additional author effort because it already understands the laser joint feature scripts the author used when creating their CAD model. Further, because metaSVG stores its additional information in a `<metadata>` tag within a standards-compliant SVG, the author is able to upload the file to a model sharing site just as if it were a plain SVG. Makers that do not use metaSVG Print are able to use the file just as if it were a plain SVG.

When the maker downloads the file, they use metaSVG Print to automatically adapt the cut plan to their desired material and thickness, and – after loading the current calibration file shared by the shop assistant – the current kerf of the makerspace's laser cutter. They take the resulting adapted path to the laser cutter and create pieces that fit well without any further iteration. This workflow serves the author, maker, and shop assistant.

# 3 RELATED WORK

Our work builds on research into improving laser cutting workflows, portability in fabrication, and encoding human expertise for reuse.

## 3.1 Improving Laser Cutting Workflows

Researchers explored several interaction concepts to improve the laser cutting process and workflow, mainly to assist users in creating designs, and in fabricating their designs. For creating designs, researchers developed fabrication-aware design tools that support users in designing models that can be fabricated out of 2D shapes, such as modeling 3D objects from 2D parts (SketchChair [35], Platener [4], LaserStacker [42], FlatFitFab [18]) or voxels (Kyub [3]). Other design tools explored parametric joint generation that account for material thickness to support users in creating designs with joints and connected parts (CutCAD [9], Joinery [48], Designosaur [23], Enclosed [44], Fresh Press Modeler [6]).

To help users fabricate their designs with laser cutters, researchers created tools that facilitate the positioning of designs on materials (Constructable [22], Sketch it Make it [14], MARCut [15], VAL [45]), that enable easier cut setting assignment for specific materials (VisiCut [26]), and that pack designs on material sheets for reducing waste (PacCAM [34], Autodesk 123D [2]), even interactively in design phase (Fabricaide [37]). Another line of research focused on extending the capabilities of laser cutters to enable users to fabricate sheets beyond 2D forms (LaserOrigami [21], LaserStacker [42], Foldem [5]) or traditional materials (BlowFab [46], Layered Fabric Printer [27], Platener [4], StackMold [43]). While these efforts address several problems and limitations, fieldwork highlights many opportunities for streamlining existing design-download-customize-print workflows. In particular, there are opportunities for supporting authors to provide richer metadata, clarifications, and expert tips to help makers successfully customize, adapt, and fabricate the designs they downloaded [1].

Our goal is to act on these opportunities. MetaSVG adds to the work on fabrication-aware design tools; however, our focus is on enabling authors to annotate metadata rather than facilitating the design process. MetaSVG Print builds on the research that facilitates the fabrication of laser cut designs; we aim to help makers successfully adapt and fabricate 2D cut plans with joints using their materials and laser cutters.

## 3.2 Portability in Fabrication

Recent HCI research raised the issue of portability in design and fabrication processes. In laser cutting, researchers investigated the machine-dependency problem due to differences in kerf. Spring-Fit [32] introduced cantilever-based springs to replace press-fit joints and mounts, and KerfCanceler [30] presented a tool to replace moving mechanisms with wedge elements, so that 2D cut plans work across laser cutters. While replacing existing joints with kerf invariant joints and mechanisms enable the portability of 2D cut plans, it does not provide an option for different types of joint fits, and it alters the design aesthetics. Kyub [3] introduced a voxel-based 3D modeling tool that performed kerf correction during 2D export, however it was limited to creating objects with only box joints, and it requires makers to have knowledge of kerf allowances. Likewise, there are several online laser cut box generators with kerf correction (MakerCase [11], Boxes.py [7]) that are subject to a similar set of limitations.

Building on Kyub, Assembler [31] infers joint and assembly metadata that is lost in traditional SVG export, enabling both path adaptation and further editing, though at the price of some user hinting. Other projects explored portability in 3D printing. Mix&Match [38] and FitMaker [16] addressed adapting downloaded 3D models to makers' tools for models to print reliably. Our work extends this line of research on portability in digital fabrication. Instead of taking a reverse engineering approach to make existing 2D cut plans

parametric and portable, we focus on creating a workflow where material- and joint-specific information can be encoded in parametric cut plans at export time, and designs are adapted for laser cutters using machine-specific profiles at print time.

## 3.3 Encoding Human Expertise for Reuse

Researchers explored how to capture and encode human expertise and design intent in fabrication in a reusable way, mostly in 3D printing. Several projects presented functional design templates and hierarchies for non-experts, based on analyzing expert-created, fabricable 3D models (PARTs [10], Design and Fabrication by Example [36]). Similarly, Grafter [33] explored how mechanical parts from different 3D models can be combined for reuse. Researchers state that for makers to reuse functional 3D designs, more accessible applications are needed to support modifying designs while conveying authors' design intent [10]. While inexperienced makers seek help and troubleshooting from shop assistants and operators, customizing designs is not an easy task for them, even when the design is parametric [10, 13].

Encoding human expertise can benefit authors as well. Recently, a study with fabrication professionals has shown that expert users desire improved fabrication workflows that have awareness of design and materials, and tools that document use settings in shared spaces [47]. They also note that shop assistants want tools that help with calibration and maintenance, as calibration of settings for specific fabrication setups can be difficult due to machine wear. Our work aligns with this vision. Our goal is to develop means for encoding and operationalizing this material- and machine-specific knowledge in a reusable way.

# 4 METASVG WORKFLOW

To address the challenge of portability and improve laser cutting workflows, we developed (1) metaSVG, a standards-compliant SVG file containing material- and machine-dependent annotations in 2D cut plans, (2) metaSVG Exporter, an OnShape extension including custom FeatureScripts for parametric joint generation and an export script for exporting metaSVG cut plans from 3D CAD, and (3) metaSVG Print, a browser-based application that takes a metaSVG file as an input and adapts the cut plans to users' materials, material thickness, scale, and laser cutters.

This section describes the format and tools in more detail from a user perspective, while the next section describes lower-level implementation details.

## 4.1 metaSVG Files Behave like SVG Files

Scalable Vector Graphics (SVG) is a graphics file format that describes elements such as lines, curves, and colors with XML text. SVG is a common exchange format for sharing laser cut designs [32], despite requiring by-hand manipulation to adapt to different materials and cutters.

Our metaSVG format extends SVG by including parametric design information in the `<metadata>` block of the file (technical details in Section 5.1). Since metadata is ignored by applications that don't understand it, users treating a metaSVG file like a regular SVG will get the behavior they expect. Users that open the file in metaSVG-aware applications will benefit from the adaptability provided by the extra information.

## 4.2 Authors Create metaSVG Files With Parametric 3D CAD

A typical authoring workflow for laser-cut items involves building the object in 3D parametric CAD and then flattening and exporting the file. One of the advantages of working in parametric CAD is that custom features – subroutines that create complex geometry, e.g., for laser-cut joints [39, 40] – can be used to ease the creation process. When creating a metaSVG file, authors use this same CAD workflow

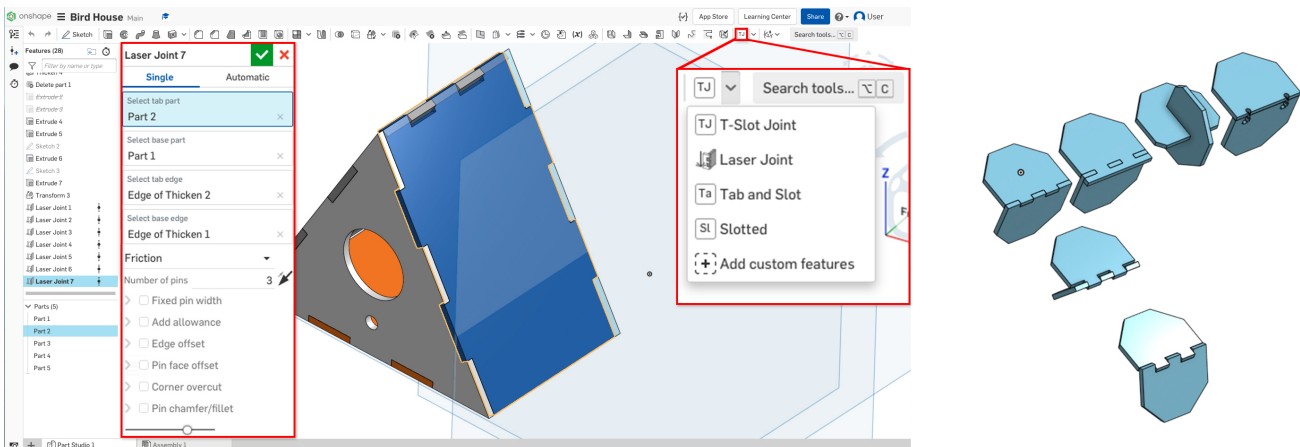

Figure 4: Authors use a parametric 3D CAD tool (OnShape) with existing laser joint features to author metaSVG files. Features are small programs used in parametric CAD to automate repetitive tasks. The list of supported metaSVG laser joint features is shown to the right, and the parameters for a box joint feature are shown to the left.

Table 1: Our metaSVG Print utility reads material, joint fit, and laser cutter settings from a "presets" table. These are the presets tables used for examples in this paper. Joint fit is documented per joint and per fit (e.g. Box-Clearance). All dimensions are in millimeters.

| Preset (ULS) | Notes | Thickness | Width | Height | Style | Kerf | BoxC | BoxL | BoxI |
|---|---|---|---|---|---|---|---|---|---|
| plywood-3mm | p:100 s:5 | 3 | 450 | 350 | #FF0000;0.001pt; | 0.05 | 0.05 | -0.05 | -0.075 |
| plywood-6mm | p:80 s:10 | 6 | 450 | 300 | #FF0000;0.001pt; | 0.05 | 0.1 | 0.05 | 0.0 |
| acrylic-3mm | p:100 s:3.5 | 3 | 450 | 300 | #FF0000;0.001pt; | 0.1 | 0.1 | 0.0 | NA |
| mdf-default | p:100 s:7 | 3 | 600 | 300 | #FF0000;0.001pt; | 0.1 | 0.1 | 0.0 | -0.07 |

| Preset (Epilog) | Notes | Thickness | Width | Height | Style | Kerf | BoxC | BoxL | BoxI |
|---|---|---|---|---|---|---|---|---|---|
| plywood-3mm | p:100 s:25 | 3 | 450 | 350 | #FF0000;0.001pt; | 0.1 | 0.05 | -0.05 | -0.075 |
| plywood-6mm | p:100 s:8 | 6 | 450 | 300 | #FF0000;0.001pt; | 0.1 | 0.1 | 0.04 | 0.0 |
| acrylic-3mm | p:100 s:12 | 3 | 450 | 300 | #FF0000;0.001pt; | 0.05 | 0.1 | 0.0 | NA |

(Figure 4), with the twist that they use slightly-modified versions of existing laser joint features that can be interpreted by our metaSVG Export script (details in Section 5.2) to produce a metaSVG file.

### 4.3 Makers Adapt metaSVG files with metaSVG Print

The process of adapting a metaSVG file to a new laser cutter, material, or scale is accomplished by a maker by using metaSVG Print (Figure 5). This dialog is modelled on a print dialog box and provides quick access to recall laser cutter and materials settings from a list of "presets". MetaSVG Print also allows makers to view and modify joint-specific parameters by clicking on either edge of a joint pair. While these adjustments are optional, they provide increased control for advanced makers.

#### 4.3.1 Shop Assistants Provide Calibration Information as Presets

The metaSVG Print dialog provides quick access to a list of "presets" – settings that the maker can select to automatically configure laser cutter kerf, joint fit, and material thickness (e.g., Table 1 shows the presets we used in producing examples for this paper). In the context of a shared-use laser cutter in a shop, we envision that a shop assistant could update these presets to reflect the shop's current laser cutter calibration and material types, allowing "one-click" access for makers.

### 5 IMPLEMENTATION

In our prototype implementation of the metaSVG workflow, we chose to build on the OnShape [24] cloud-based CAD system for authoring, with custom FeatureScripts [25] for joint description, and

a python script for exporting. For our metaSVG Print utility we used a browser and javascript-based UI with a python backend. In this section, we describe our system architecture (Figure 6), file format, and implementation in more detail. The code and documentation are available on the project website [1].

### 5.1 The metaSVG Format

MetaSVG files contain a model representation of a cut plan that annotates material- and joint-specific metadata into a standard SVG with a `<metadata>` tag (Figure 7). In this metadata section, metaSVG stores a hierarchy of information rooted at faces and joints. Faces represent an individual planar part to be cut. They are defined by an outer perimeter path, and can have any number of cut paths within this perimeter. Joints represent how these faces come together. Each joint references two mating edges and a set of joint-specific parameters (Figure 9). Together, the metadata in a metaSVG is sufficient to completely re-compute the path information stored in the traditional SVG part of the file.

### 5.2 The metaSVG Exporter

In our prototype implementation, the process of creating a metaSVG file from a parametric 3D CAD model is carried out by a python script that accesses OnShape's REST API. This python script uses the OnShape API to read data from modified FeatureScripts (OnShape's name for code that implements custom features – in this case, for laser joint generation), as well as to read general shape information from the CAD model.

---

[1] http://graphics.cs.cmu.edu/projects/metasvg/

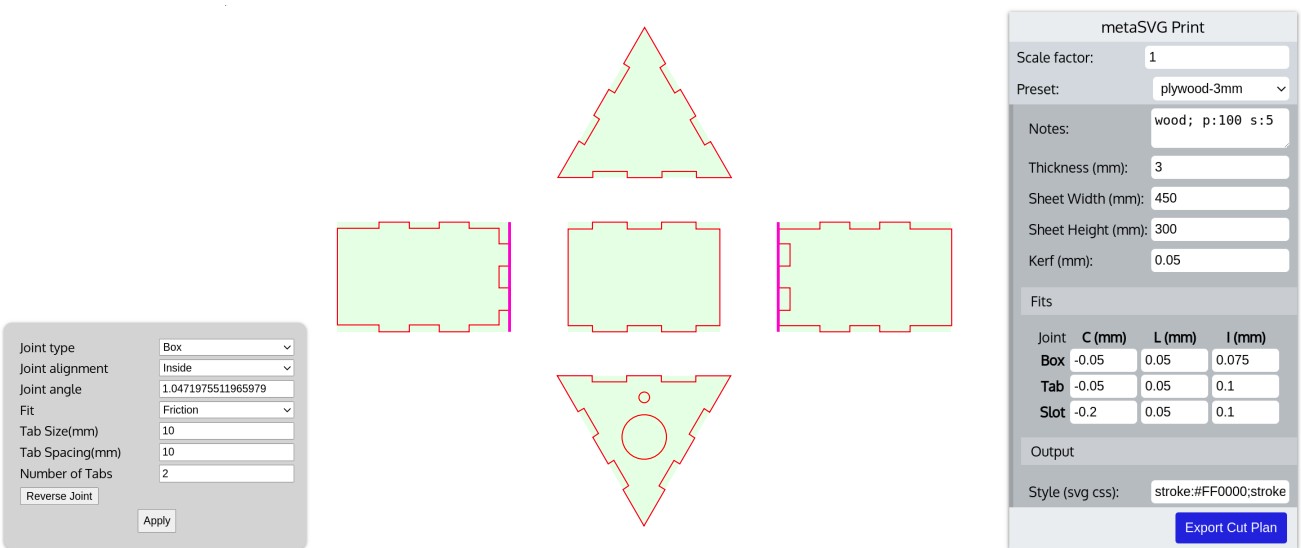

Figure 5: The metaSVG Print dialog allows makers to select model and laser cutter settings (with presets available for common cases). A "cut preview" to the left shows the adapted plan. Advanced users can click on the cut preview to read and customize individual joint settings.

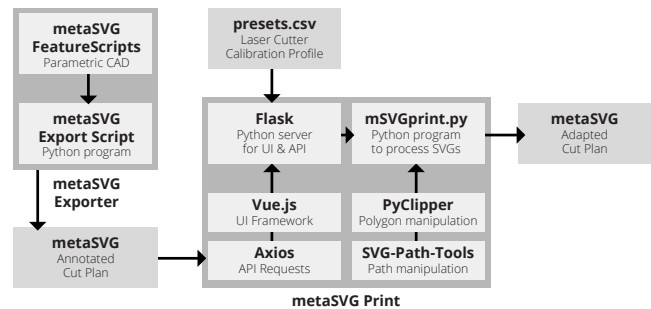

Figure 6: Data pipeline for metaSVG Export and metaSVG Print: Our system inputs, adapts, and outputs metaSVG cut plans.

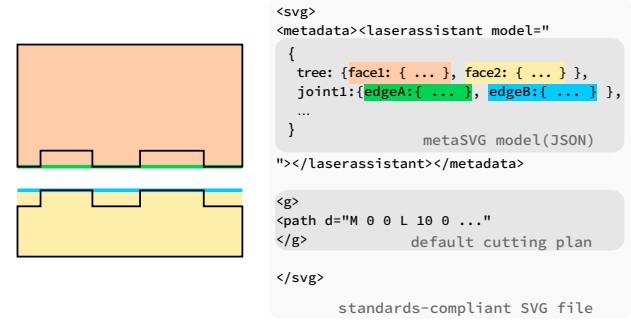

Figure 7: A metaSVG file contains both a regular cut plan stored in SVG format (black) and embedded JSON metadata containing a parametric version of the plan (colorful highlights). This parametric data allows the cut plan to be recomputed for new materials, thicknesses, scales, and laser cutters.

### 5.2.1 metaSVG FeatureScripts

MetaSVG currently supports four commonly used laser cut joint types (Figure 8): (a) box joints (aka finger joints), (b) tab-and-slot joints (aka mortise and tenon joints), (c) slotted joints (aka cross joints or interlocking joints), and T-slot joints (aka captive nut and screw joints). All joints, except the slotted joint, build on and extend two existing custom features: the Laser Joint [39] and T-Slot [40] FeatureScripts. Our modifications to these scripts are relatively minor, and include asking for joint fit information and some additional metadata to assist our script in computing joint locations. Joints are created between two planar parts, and the joint location and angle are computed by using the largest face's normal from either part to define the plane of the parts. Figure 9 shows the joint parameters stored by each joint.

### 5.2.2 metaSVG Export

Our metaSVG export script uses OnShape API to retrieve a list of parts and a list of joints. It then matches the joints with edges to create the metaSVG file. To do this, it tells OnShape to "suppress" the various laser joint features (that is, to ignore them when computing the final shape of the object). This allows our script to retrieve the shape of each part of the model without the extra serrations and tabs from the joints. From each of these parts it selects one of the largest faces (for a general planar part there will be two equal largest faces) and uses this face's normal to define the orientation of the part and compute outer and inner 2D cut paths for the part. The script then loops through each laser joint feature and associates the feature with the parts by looking up the edge IDs stored in the feature's parameters; and, further, extracts various joint parameters required for each joint type (Figure 9). Finally, the program un-suppresses the joint features and flattens the model again in order to write out the standard SVG portion of the file.

### 5.3 metaSVG Print

MetaSVG Print extracts the metadata from a metaSVG file and uses it (along with a preset defining laser cutter and material parameters) to generate a cut plan. MetaSVG Print consists of a browser interface and a python backend.

We also developed a command-line version of metaSVG Print which has the same functionality but doesn't require any GUI interaction. This version makes it easy to batch process metaSVG files.

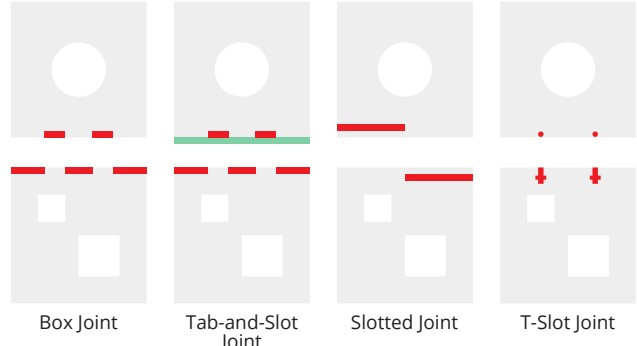

Box Joint  Tab-and-Slot Joint  Slotted Joint  T-Slot Joint

Figure 8: MetaSVG Print expresses joints in terms of regions of material that must be added and subtracted in order to realize the joints. When preparing an output, first the areas of addition from all joints (green) are added, followed by subtracting the areas of subtraction (red) from all joints.

### 5.3.1 Frontend

The frontend is a web interface made with Vue framework and uses Axios to make calls to the backend via Flask. The frontend is a relatively thin wrapper around the backend code, and provides only basic display and parameter setting functionality. The frontend aids in parameter setting by providing the user with a list of presets, which it fetches from the backend to allow for easy setting of cutter- and material- specific settings. Any edits to global parameters or per-joint settings are sent to the backend via HTTP request, to which the backend responds with an updated metaSVG whose SVG portion has been replaced with the results of regenerating the parts with the given parameters.

### 5.3.2 Backend

The backend processes the cut plans (Figure 10) using two main libraries: SVG-Path-Tools [28] and PyClipper [29]. SVG-Path-Tools is used to read the metaSVG metadata. Next, faces and cutouts are regenerated as outer and inner loops using the PyClipper library. Joints are regenerated as SVG path strings through straightforward arithmetic and parsing of joint-specific parameters; specifically, each joint defines both "positive" areas to be added to the model and a "negative" areas to be cut from the model. These joint areas aligned to edges using SVG-Path-Tools, and then merged with faces using PyClipper. The resulting geometry represents the cut plan without allowances and line formatting. Next, the cut plan is offset to account for kerf based on the value given in the laser cutter profile. Finally, using the standard python XML library ElementTree, the paths are formatted into a standard SVG with the metaSVG data embedded within a `<metadata>` tag.

### 5.3.3 Joint Thickness

When box joints don't meet at a 90-degree angle, the size of tabs need to be adjusted to account for this. Our code computes the angle-adjusted joint width, $t'$, from the material thickness, $t$, and the joint angle, $\theta$, as follows:

$$t' = \begin{cases} t\tan(\pi/2 - \theta) + \frac{t}{\cos(\pi/2-\theta)} & \text{if } \theta < \frac{\pi}{2} \\ t\sin(\theta) & \text{otherwise} \end{cases}$$

(Our code currently only performs this compensation for box joints, though similar formulae could likely be derived for tab-and-slot joints as well.)

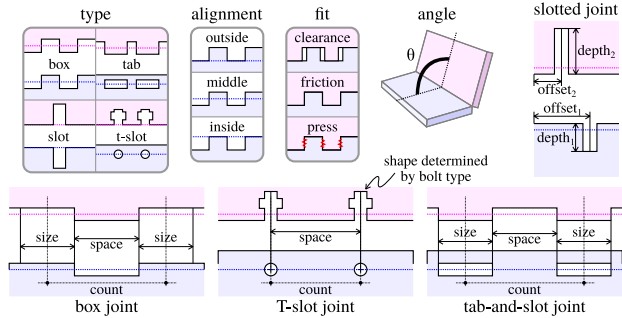

Figure 9: Joint parameters used in metaSVG. All joints store their type, alignment, fit, and angle. Individual joint types use additional parameters as shown. Dimensions are shown before adjusting for fit. Tab depth for box and tab-and-slot joints and slot width for slotted joints are computed from material thickness and joint angle (see paper text for details).

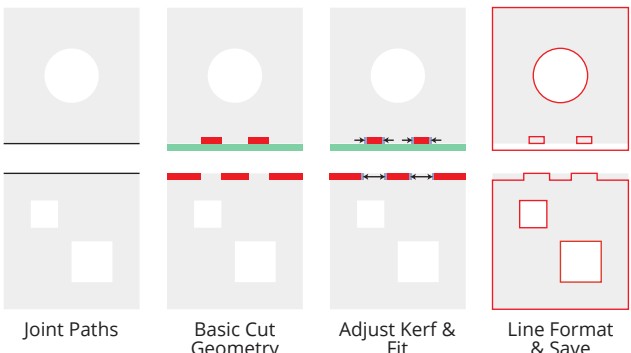

Joint Paths  Basic Cut Geometry  Adjust Kerf & Fit  Line Format & Save

Figure 10: metaSVG Print processing steps to regenerate joints.

### 5.3.4 Joint Fit Compensation

Joint fit is the planned allowance between two mating parts in laser cut joints. It is used to determine how loosely or tightly the parts fit together. MetaSVG supports three kinds of joint fit based on an engineering classification: clearance fit (aka loose fit), location fit (aka friction fit), and interference fit (aka press or hammer fit) [20]. When modeling in OnShape, MetaSVG FeatureScripts contain a joint fit drop-down, with location fit as the default choice.

The joint fit allowance is material-, joint-, and machine-dependent. For this reason, expert users typically calibrate the joint fit by offsetting cut paths in small increments until an offset provides the desired fit [19].

MetaSVG Print takes offsets for each of the fit types as part of its parameters. MetaSVG Print uses these "fit offsets" to adjust the joint features in a joint-specific fashion: for box joints, the fit offsets adjust the size of tabs; for tab-and-slot joints, the adjust the size of tabs and slots (in addition, the thickness of slots is always adjusted by the clearance fit offset); and for slotted joints the fit offset is applied to the slot thickness. The fit offset is not used for t-slot joints, since the bolts serve to adjust fit tightness.

### 5.3.5 Presets

All of the parameters used by metaSVG Print (other than the design scale) are specific to a given laser cutter, cut setting, and material type. As such, we make it easy to select them in a batch by using a "preset". Presets are stored in a `presets.csv` (comma-separated value) table, which is easy to update when laser cutter performance changes over time, or when moving to a new lab with a different laser cutter.

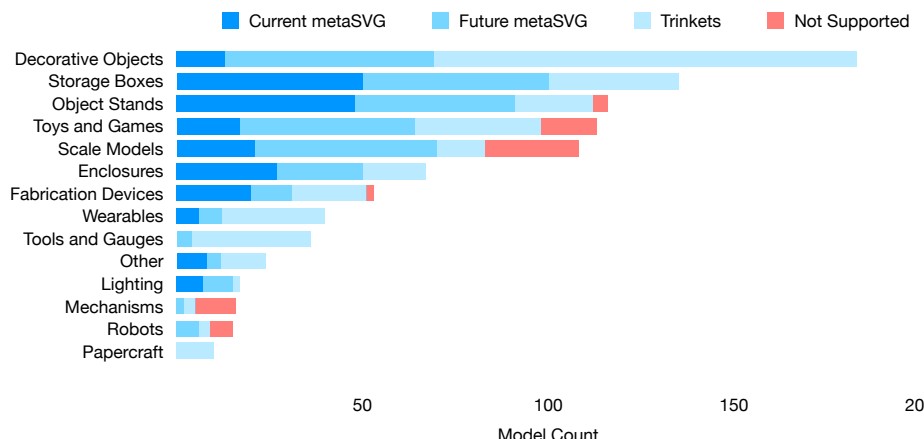

Figure 11: Laser cut plans on Thingiverse by category and applicability of metaSVG. "Current metaSVG" (23.2%) indicates cut plans that should be supported by our research prototype; "future metaSVG" (33.1%) are cut plans that would work with our prototype if additional joint types are added; "trinkets" (36.8%) are cut plans which derive no benefit from metaSVG (i.e., because they have no joints); and "not Supported" (6.9%) models do not fit the assumptions of metaSVG.

Table 2: Models we recreated for evaluation. Numbers in parentheses denote the number of edges that could not be adapted successfully.

| Model | Category | Thing ID | Adaptation | Box | Tab-Slot | Slotted | T-slot |
|---|---|---|---|---|---|---|---|
| 3D Printer Frame | Fabrication Devices | 8563 | Laser cutter | - | 6 | - | 12 |
| VR Goggles | Wearables | 638605 | Laser cutter | 4 | - | 6 (4) | - |
| Candle Holder | Lighting | 2729105 | Thickness | 24 | - | - | - |
| Raspberry Pi Case | Enclosures | 101946 | Material, Thickness | 4 | 4 | - | 4 |
| Arcade Cabinet | Enclosures | 1428410 | Material, Thickness | 12 (1) | - | - | 8 |
| Laptop Stand | Object Stands | 3199311 | Material, Thickness | 2 | 2 | - | 2 |
| Tray | Storage Boxes | 2735730 | Laser cutter | 8 | - | 10 (4) | - |
| House | Scale Models | 916049 | Scale | 5 | - | - | - |
| Dice Tower | Toys and Games | 3954998 | Material | 17 | - | 8 (6) | - |
| Windmill | Decorative Objects | 2287464 | Thickness | 56 | - | 8 | - |
| Chair | Other | 12037 | Scale | 9 | - | - | - |
| Bird House | Other | 1691891 | Scale | 7 | - | - | - |

In order to make it easier to dial in new settings, metaSVG Print does allow settings to be edited after a preset is loaded, though these edits are not preserved unless the user manually adds them to the presets file.

## 6 TECHNICAL EVALUATION

To assess the performance of our system, we tested it with 12 models. We selected these models based on an analysis of laser cut plans in Thingiverse. First, we searched for "lasercut" models, and filtered models that have "makes" by other users to ensure they could be successfully fabricated. Our survey returned 1107 results. Next, we eliminated non-laser cuttable models (174), and we conducted affinity diagramming [12] to group cut plans. Our analysis revealed 14 categories (Figure 11).

To better understand the coverage, we assessed whether the models in each group could be fabricated using the metaSVG file format and workflow. We reviewed the photographs and cut plans for each model to identify the types of joints used. This resulted in a mapping of the design space into four groups: (1) Current metaSVG, the models that our system should be able to support, (2) Future metaSVG, the models that could be supported by extending the current joint library (e.g., living hinges, stacked planes, models with varying material thicknesses, joints connecting to more than a single other joint), (3) Trinkets, models where precision might not be critical, and (4) models that cannot be supported (Figure 11). We targeted models in current metaSVG group (23.2% of all models, 36.8% if trinkets excluded) for our technical evaluation, while our system can potentially address all groups except the last (6.9% in total, 10.8% if trinkets excluded).

From *current metaSVG* models of each category, we selected one or two models that were representative of the category (Table 2). We excluded the categories robots, mechanisms, tools and gauges, and papercraft as they did not have any current metaSVG models. To test our authoring workflow, we rebuilt the models in OnShape using metaSVG FeatureScripts and exported them as metaSVGs. To test our adaptation workflow, we adapted them for fabrication using metaSVG Print. We tested adaptation across materials (3mm and 6mm plywood, and 3mm acrylic), laser cutters (a ULS 60 watt laser cutter with 0.05mm kerf and an Epilog Mini 50 watt laser cutter with 0.1mm kerf), and scales.

### 6.1 Results

Ten of the twelve models were readily adaptable. Of these ten models, Arcade Cabinet and Tray had some joints that did not adapt due to implementation bugs and required manual adjustment. The remaining two models (VR Goggles and Dice Tower) posed challenges due to having variations of tab-slot and slotted joints that are not supported by our current system. However, the remaining joints were adapted; all models were supported by our workflow. Figure 12 shows some of the models we adapted and fabricated, demonstrating material, thickness, scale, and laser cutter change. For models adapted to be cut out of acrylic, thickness also had to

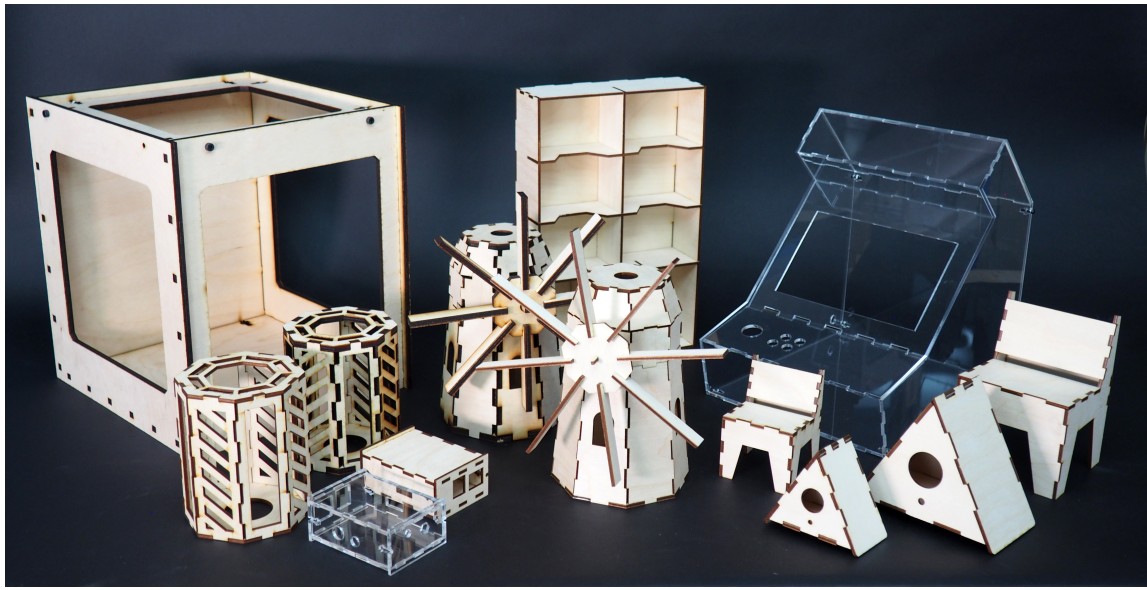

Figure 12: Some of the fabricated models showing adaptations to different materials, material thicknesses, scales, and laser cutters.

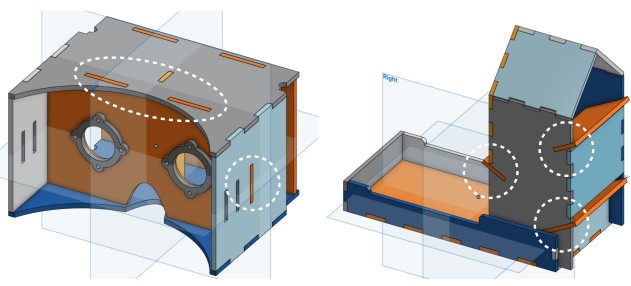

Figure 13: Models including joint variations (tabs in the center of faces, angled slots) that are not supported by our current metaSVG implementation. We plan to add support for more joint types and variations in the future.

be adapted to compensate for same-material variations (e.g., 3mm nominal thickness often measured 2.7-2.9mm actual thickness). All adapted models assembled successfully, except for Windmill, which had a square shaft and shaft cutout that required manual adjustment after thickness change.

## 7   DISCUSSION, LIMITATIONS, AND FUTURE WORK

We view metaSVG, and the workflow we have outlined above, as a way to retrofit the current laser cutting plan ecosystem in a way that benefits all involved roles – authors, makers, and shop assistants – without requiring any additional work.

The core idea of metaSVG is to capture the data that is required to make a laser cutting plan adaptable by augmenting authoring tools. A natural question to ask is whether metaSVG captures the right amount of information. For example, one could imagine a "heavier" metaSVG which holds the full CAD file or a "lighter" metaSVG which contains a few variations of the cut plan and handles all parameter changes through interpolation. We would argue that metaSVG contains just enough information to support the maker's workflow. Though we do think it is interesting to consider the ways in which parametric CAD documents are – in essence – small geometric programs for which one might want to construct alternative (more limited) interfaces for specific users.

Our current system is a research prototype and requires a fair bit of polishing before it can be considered end-user ready. Particularly, while our system is already able to generate a wide range of results, several refinements will need to be made to the system before it is ready for wide release.

**Extra parameters.** Isolating the particular edge of a face along which a joint should be created requires some tricky geometric computations. Our prototype scripts avoid these computations by requiring extra (redundant) selections of edge(s) and face(s) along with parts. This can be seen in Figure 4. In the future, we plan to revisit the computations to eliminate these redundant parameters.

**Joint types and variations.** Our system supports box, tab-and-slot, t-slot, and slotted joint types; however, it does not support all variations of these joints. In the future, we plan to expand the list of supported joints and variations; of particular priority for us are tab-and-slot joints that are not located near edges and slotted joints with non-90-degree angles between the pieces – since these joint types are used in many existing laser-cut objects.

**OnShape quirks.** The IDs that OnShape assigns to parts and features change when FeatureScripts are enabled and disabled. This means that occasionally it can be difficult for our processing script to figure out where (on a "clean" model) each joint is located; requiring the joints to be re-ordered in the CAD model before an export succeeds. In the future, we plan to modify our metaSVG export script to use geometric information (i.e., 3D positions and normals) to make these correspondences, rather than IDs. Further, OnShape's notion of face and part orientation seems somewhat arbitrary, which can also lead to processing quirks that must be resolved by selecting the other side of an object. We plan to extend our script to compute its own inside/outside orientations to avoid these quirks.

**Laser Cutter quirks.** In metaSVG, "kerf" [cut width] is the interface between human preference (e.g., for tightness of joint fit) and laser cutter behavior (e.g., the cut settings used). That is, metaSVG processors (like many human laser cutter users) treat this single number as capturing everything there is to know about the mechanical behavior of a laser cut. Unfortunately, our experience in setting up fit value tables reveals that this is an often-insufficient approximation. (This is why metaSVG uses per-preset corrections to its fit values, rather than per-material corrections.) In practice, the mechanical behavior of cuts varies quite significantly depending

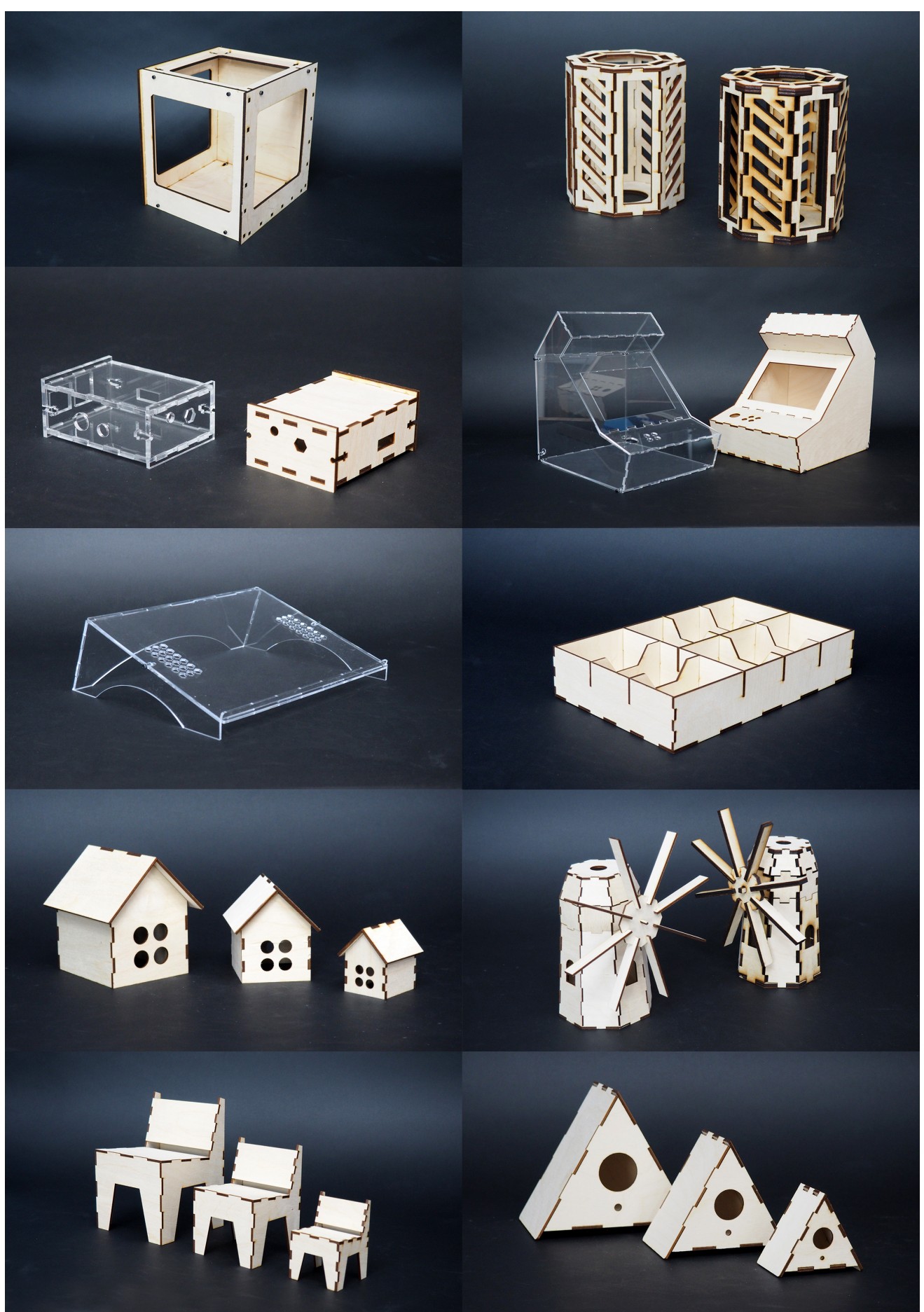

Figure 14: Detailed views of fabricated models.

on how they are formed; we conjecture that this is both because of depth-dependent cut width variation and the level of combustion products left at the edge. In the future, we think it would be exciting to generalize kerf to a multidimensional quantity that actually *does* capture the general mechanical behavior of cuts. This would allow users to specify their preferred fit values once and avoid needing to correct them, while a "smart" laser cutter could use its tracked multi-dimensional kerf value to adapt users fit values to any material and cut setting.

And, indeed, this is what we see as the core idea of metaSVG: it brings together human knowledge with machine understanding.

## 8 CONCLUSION

We presented a portable file format and an accelerated workflow for exchanging adaptable laser cutting plans. Our assessment demonstrated that the metaSVG format is capable of representing typical laser cut designs that are commonly shared online. We envision a future where one can send a metaSVG file to a laser cutter icon with at least as much confidence in the output fidelity as one has when sending a postscript file to a 2D printer. This will, require further integration of our system with laser cutting hardware and software to prototype a seamless design, calibration, and fabrication workflow enhanced with sensing.

More broadly, we observe that metaSVG is addressing the problems created when a workflow fails to capture relevant, expert-provided knowledge. We observe that many fabrication workflows beyond laser cutting suffer from this problem – from 3D models "forgetting" that a certain hole is a bolt hole and thus shouldn't be scaled, to 2D images "forgetting" layers and grouping behaviors that make them more easily editable. Often, research effort is expended trying to hallucinate and retrofit this missing knowledge into existing files; we encourage researchers to also consider building tools that avoid throwing out such information in the first place.

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
