# OpenReview forum: "metaSVG: A Portable Exchange Format for Adaptable Laser Cutting Plans"
_graphicsinterface.org/Graphics_Interface/2022/Conference — GI 2022_

### Official Review · Reviewer_Xret · 2022-04-08
**Excellent concept for improving portability of laser cutting plans through SVG metadata, but lacking reproducibility.**

**Rating:** 6
**Confidence:** 4

**Review:**

Originality & Significance:
The submission presents an original and highly practical new workflow for portability of laser cutting plans, termed metaSVG. The core contribution is to leverage metadata tags in standard SVG files to embed information on joints, which can be used to automatically adapt a cut plan to new materials, material thicknesses, model sizes, and machine-specific settings. The paper presents an end-to-end workflow, including a metaSVG exporter, and metaSVG print utility that respectively created the annotated cut plan and applies the adaptations.

Clarity:
The submission is clearly written and easy to understand. Some minor comments that might improve exposition:
- The paper could be shortened. Sec. 2 Motivation, while excellent, does not need to be so detailed. Figs. 12 and 14 are redundant.
- I found the journey maps in Fig. 2 a bit confusing. Are the pail yellow connections and emotion lines directional? Some emotions appear to start and end at phases in the workflow and others do not.
- Fig 9. include the depth parameter for the tab-and-slot joint.
- Table 1: please explain the table contents, particular BoxC, BoxL, BoxI.

Reproducibility:
The weakness of this submission is lack of reproducibility. There is very little information given about the metaSVG format itself, which makes it difficult to evaluate and for others to reproduce or build on the work. It appears the only documentation given is Sec. 5.1 (1 paragraph) and Fig. 7 which shows a very sparse skeleton. The format should be a major component of the paper. Show the complete contents of the "tree" and "joint" elements. Demonstrate multiple joints on a face, the various joint types, how joint parameters are represented, etc. It is not clear to me how joints and faces in the metadata refer to elements in the standard SVG file. It would also be helpful to include sample calculations for computing the offsets for one or two joint types. Are joints processed independently of each other? If joints exist on adjacent edges are there any issues with compatibility?

Evaluation:
The workflow is nicely evaluated. Results of adaptation are shown for 10 models. Applicability is also justified by a survey of models on Thingiverse. A selection of 1107 models were assessed (was this done manually?) and found that 23.2% could be supported by metaSVG in its current implementation. A minor point, "trinkets" could have a more descriptive label.

Conclusion:
Overall I am enthusiastic about the potential contributions of this submission, but hesitant to accept with the lack of reproducibility. Far more documentation is needed of the metaSVG format.

---

### Official Review · Reviewer_5zuE · 2022-04-12
**Interesting problem but the solution is not convincing enough**

**Rating:** 5
**Confidence:** 2

**Review:**

This paper presents a novel idea to embed metadata into the SVG file such that the laser cutting plan is editable for various thicknesses of the sheet materials.

The paper has a very good motivation. While many 3D printed model is distributed using a CAD model (e.g., OpenSCAD) allowing the adjusting dimensions, it is difficult to do so in the vector graphics.

On the other hand, if the parametric CAD model is available, it is probably better to distribute the model itself. This is because the CAD model is much easy to make bigger changes including topological changes. A CAD model file and metaSVG are not very different in terms that it needs specialized software to edit the parameters. The user of the metaSVG has to create a 3D CAD model anyway using a CAD system, there is no reason the user has to distribute it in the SVG format.

The detail of the file format of the metaSVG is not discussed in the paper. It is true that the <metadata> tag allows putting any data inside SVG, it would be nice to discuss the data structure allowing the parametric edit. This description is important because the paper tries to make the metaSVG as portable as possible.

a typo in Section 7, "away" should be "a way"

Overall, this paper tries to solve an interesting problem. However, the paper is not yet mature enough.

---

### Official Review · Reviewer_nit1 · 2022-04-13
**Good contribution with a possible important impact**

**Rating:** 7
**Confidence:** 3

**Review:**

The paper presents a new format, metaSVG that extends SVG with laser cutting--specific features, an exporter for that format, and an importer/'print' application that can load a laser cutter-specific profile. They validate it on a variety of models from Thingiverse that they exported to metaSVG and tested on a number of materials and laser cutters.

I think this paper is a great contribution, addressing the notorious problem of portability/compatibility of laser cutting designs. It is well written, well validated, and discusses its limitations well. I think it will have an important impact on the fabrication community.

My main hope is that the authors will publish the code, documentation, the exporter, etc. for public use. My only concern is how easy it is to create the calibration profiles for a maker, I think that might be the limiting factor to wide spread of metaSVG. My guess is that it might require an extra interface or at least very detailed and easy guidelines.

---

### Decision · Program_Chairs · 2022-04-17

Accept